# A New Approach to Border Irregularity Assessment with Application in Skin Pathology

**Pawel Kleczek** [1,*] **Grzegorz Dyduch** [2], **Agnieszka Graczyk-Jarzynka** [3]
**and Joanna Jaworek-Korjakowska** [1]

[1] Department of Automatic Control and Robotics, AGH University of Science and Technology,
al. A. Mickiewicza 30, 30-059 Krakow, Poland; jaworek@agh.edu.pl

[2] Chair of Pathomorphology, Jagiellonian University Medical College, ul. Grzegorzecka 16,
31-531 Krakow, Poland; grzegorz.dyduch@cm-uj.krakow.pl

[3] Department of Immunology, Medical University of Warsaw, ul. Nielubowicza 5, 02-097 Warsaw, Poland;
agnieszka.graczyk-jarzynka@wum.edu.pl

* Correspondence: pkleczek@agh.edu.pl

**Abstract:** The border irregularity assessment of tissue structures is an important step in medical diagnostics (e.g., in dermatoscopy, pathology, and cardiology). The diagnostic criteria based on the degree of uniformity and symmetry of border irregularities are particularly vital in dermatopathology, to distinguish between benign and malignant skin lesions. We propose a new method for the segmentation of individual border projections and measuring their morphometry. It is based mainly on analyzing the curvature of the object's border to identify endpoints of projection bases, and on analyzing object's skeleton in the graph representation to identify bases of projections and their location along the object's main axis. The proposed segmentation method has been tested on 25 skin whole slide images of common melanocytic lesions. In total, 825 out of 992 (83%) manually segmented retes (projections of epidermis) were detected correctly and the Jaccard similarity coefficient for the task of detecting retes was 0.798. Experimental results verified the effectiveness of the proposed approach. Our method is particularly well suited for assessing the border irregularity of human epidermis and thus could help develop computer-aided diagnostic algorithms for skin cancer detection.

**Keywords:** image analysis; border irregularity; feature detection; pathology; epidermis; morphometry; skin

## 1. Introduction

Over the past several decades, there has been a significant increase in the incidence rate and mortality caused by skin cutaneous melanoma, the most aggressive and dangerous skin cancer, among Caucasian populations worldwide [1]. Nowadays, melanoma is responsible for about 70% of skin cancer-related deaths in the United States and in Australia [2,3]. Since no effective treatment of melanoma in advanced stages has been developed so far, its early diagnosis has become an extremely important issue. When detected early, melanoma is treatable in nearly all cases with a simple surgical excision [4].

The gold standard in skin melanoma diagnosis is the histopathological examination—the microscopic examination of tissue in order to study the manifestations of disease [5]. Other forms of examination, such as dermatoscopy, are useful for screening, but their diagnostic confidence is inferior to the histopathological examination. The epidermal area (Figure 1) is a particularly important target of examination when diagnosing skin conditions, especially for nearly all types of cancer. Its morphometric and cytologic features are key factors considered when grading a skin tissue [6,7].

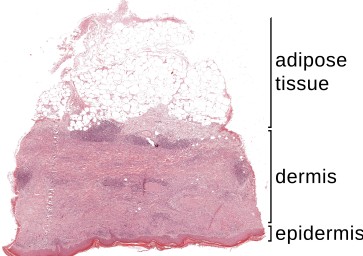

**Figure 1.** A whole slide image (WSI) of a typical skin preparation contains three main tissue layers: epidermis, dermis and adipose tissue.

The histopathological criteria currently used in the diagnosis of melanoma consist of the analysis of numerous features, such as: lesion's asymmetry, morphometric features of epidermis, proliferation patterns of single melanocytes, cytological atypia, mitoses, and necrosis [7,8]. The main criteria related to the morphometry of epidermal component are: uniformity and symmetry of epidermal thickness along the lesion, and elongation and thickening of rete ridges [7,8]. Rete ridges (retes) are downward projections of the epidermis between the underlying connective tissue (Figure 2). The epidermal thickness is measuring along its sections between bases of adjacent retes, orthogonally to the main axis of the epidermal base. The epidermis main axis is a centerline of the epidermis base, i.e., the epidermal region with retes passed over. Figure 3 shows a comparison of a typical epidermal morphometry in benign and malignant lesions.

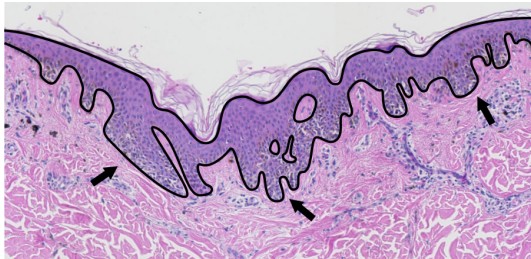

**Figure 2.** The epidermis (marked with the solid line) may have a non-trivial morphometry, with a number of rete ridges (some of them are marked with an arrow).

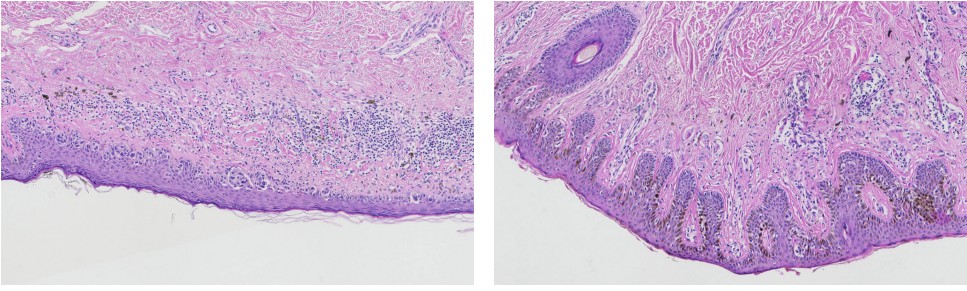

(**a**) Melanoma in situ (malignant)　　　　(**b**) Junctional dysplastic nevus (benign)

**Figure 3.** Structure comparison of the epidermal component: (**a**) melanoma in situ—effacement of rete ridges, irregular epidermal thickness; (**b**) junctional dysplastic nevus—uniformly elongated rete ridges, regular epidermal thickness.

Although traditionally pathologists examined histopathological slides under a light microscope, quantitative analysis of a large volume of specimen is a laborious and difficult task. Moreover, since pathologists make the diagnosis based on their personal clinical experience, it is often subjective and leads to intra- and inter-observer variability [9,10]. These issues may be addressed by developing automatic image analysis methods, which will provide reliable and reproducible results.

In our study we propose a novel method for automatic determination of the location, base width, length, and height of border irregularities (projections) and demonstrate its application to the detection of retes in a segmented epidermis. These information can be used to develop complex indexes describing the epidermal morphometry, which would help improve the accuracy of diagnostic algorithms for skin conditions (e.g., using cognitive analysis techniques [11,12]).

The application of our algorithm is not limited to the field of skin pathology, it can easily be adjusted and used in pathological diagnostics of organs covered with the mucous membrane (e.g., gallbladder, intestine), in dermatoscopy (to assess border irregularity of melanocytic lesions [13]—a jagged lesion border is the hallmark of melanoma), as well as in cardiology (e.g., intravascular ultrasound (IVUS) contour detection for the diagnosis of coronary atherosclerosis [14], optical coherence tomography (OCT) image analysis for edge restenosis after stent implantation [15,16], lumen segmentation method for intracoronary OCT [17,18]). To the best of our knowledge it is the first method to measure morphometric parameters of individual border projections as well as the first one to show its application to analyze the epidermal morphometry in skin histopathological images.

The main contributions of this paper are as follows:

- we propose a new border irregularity assessment method which can be used in different applications, and
- we present an accurate algorithm for the detection of individual projections and for measuring their morphometry.

*Related Works*

There are relatively few works in the literature which cover skin histopathological images processing of skin whole slide biopsy images stained with hematoxylin and eosin (H&E), the standard stain in histopathology. We list some of the notable examples and briefly detail methods for automatic diagnosis of skin lesions.

Skin epidermis segmentation problem is addressed in [19–22] using a variety of techniques, such as Otsu's thresholding, shape features, statistical analysis, stain unmixing, morphological processing, and thickness measurement. The problem of melanocytic tumor depth measurement was first addressed in [23] and then improved in [24]. Based on the work in [25] for melanocyte detection and the approach in [22] for epidermis segmentation, Xu et al. [26] proposed a computer-aided technique for automated analysis and classification of melanocytic tumor. They first segment both epidermis and dermis regions, then compute a set of features reflecting nuclear morphology and spatial distribution, and finally classify the lesion image into different categories such as melanoma, nevus or normal tissue by using a multi-class support vector machine (mSVM) with extracted epidermis and dermis features. Olsen et al. [27] used a deep learning approach to diagnose three common skin conditions—basal cell carcinomas (BCCs), dermal nevi, and seborrheic keratoses. Noroozi and Zakerolhosseini [28] proposed an automated method for differential diagnosis of squamous cell carcinoma (SCC) in situ from actinic keratosis. They first segment epidermis and remove cornified layer, then specify epidermis axis using the paths in its skeleton and remove the granular layer via connected components analysis, and finally perform classification based on intensity profiles extracted from lines perpendicular to the epidermis axis.

None of the above-mentioned image classification methods considers morphometric features of rete ridges, whereas uniformity and symmetry of both elongation and thickening of rete ridges are among most important diagnostic criteria related to the morphometry of epidermal component [7,8].

There are many works in the field of medical image processing regarding automatic diagnosis of melanocytic nevi using clinical and dermoscopy images (for example, see [29–31]) and in recent years the deep learning approach helped to significantly increase the accuracy of methods for automatic classification of melanocytic lesions [32–34]. However, to the best of our knowledge, automatic diagnosis using histopathological images has not yet been sufficiently addressed.

Since histopathological image analysis is the gold standard for diagnosing and grading skin tissue malignancies, our method will provide a valuable contribution towards automating this procedure.

## 2. Material and Methods

Image segmentation is one of the most important tasks in medical image analysis and is often the first and the most critical step in many clinical applications. Various approaches to segmentation of medical images are described in [35–37]. Although our method uses some of image segmentation techniques, the most important task in the presented method is to determine bases of rete ridges (Figure 4).

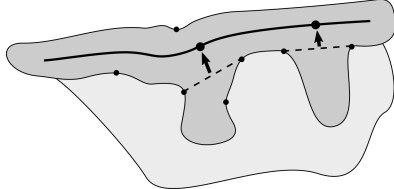

**Figure 4.** The epidermal region (dark gray) and its features: the epidermis main axis is marked with the thick solid line, the bases of retes are marked with dashed lines, their corresponding roots as points on the epidermis main axis, and candidates for rete bases as points on the epidermis border.

The "roots" of retes (i.e., the approximated location of retes along the epidermis main axis—a centerline of the epidermal region with retes passed over) are determined by analyzing the skeleton of the segmented epidermal region and finding projections branching off towards the underlying skin layers, i.e., not towards the epidermis "outer" edge (Figure 7e,f). The epidermis "outer" edge is determined mainly by analyzing the distance to the slide background along the normal (in both directions) for each point of the epidermis border, followed by morphological processing. To determine the epidermis main axis we find such a path between two most distant endpoints in the epidermis skeleton which runs closest to the epidermis "outer" edge. Endpoints forming the bases are determined by firstly analyzing the curvature of the epidermis border to identify endpoint candidates and then matching those candidate nodes with rete roots according to geometric criteria. A post-processing is usually necessary, as some automatically segmented retes contain multiple individual tips whereas others are partially joined with their neighboring retes—in such cases it is necessary to merge or split, respectively, the segmented retes. After retes are delimited, we may retouch the epidermis main axis so that it runs only through the epidermis base and not through upper parts of retes, which results in increased accuracy of locating rete roots (Figure 7h). After the aforementioned steps are carried out, we measure the morphometry of individual retes—their width, length and height (Figure 5). The flowchart in Figure 6 presents the pipeline of our method. The results of each step are shown in Figure 7.

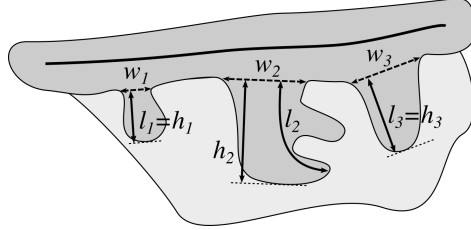

**Figure 5.** The morphometric features of projections (rete ridges): base width ($w$), length ($l$), and height ($h$).

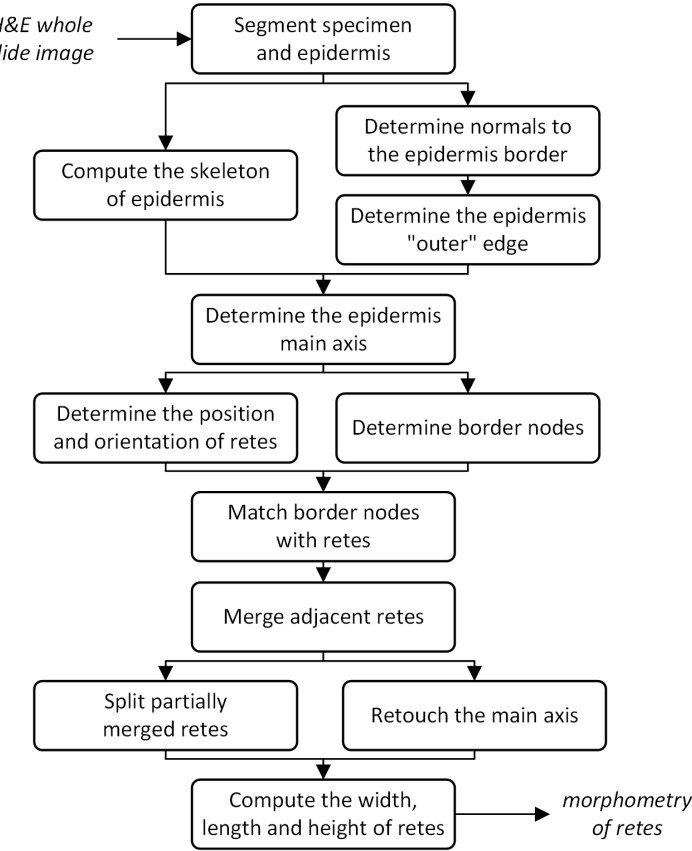

**Figure 6.** Flowchart of the pipeline of the proposed method.

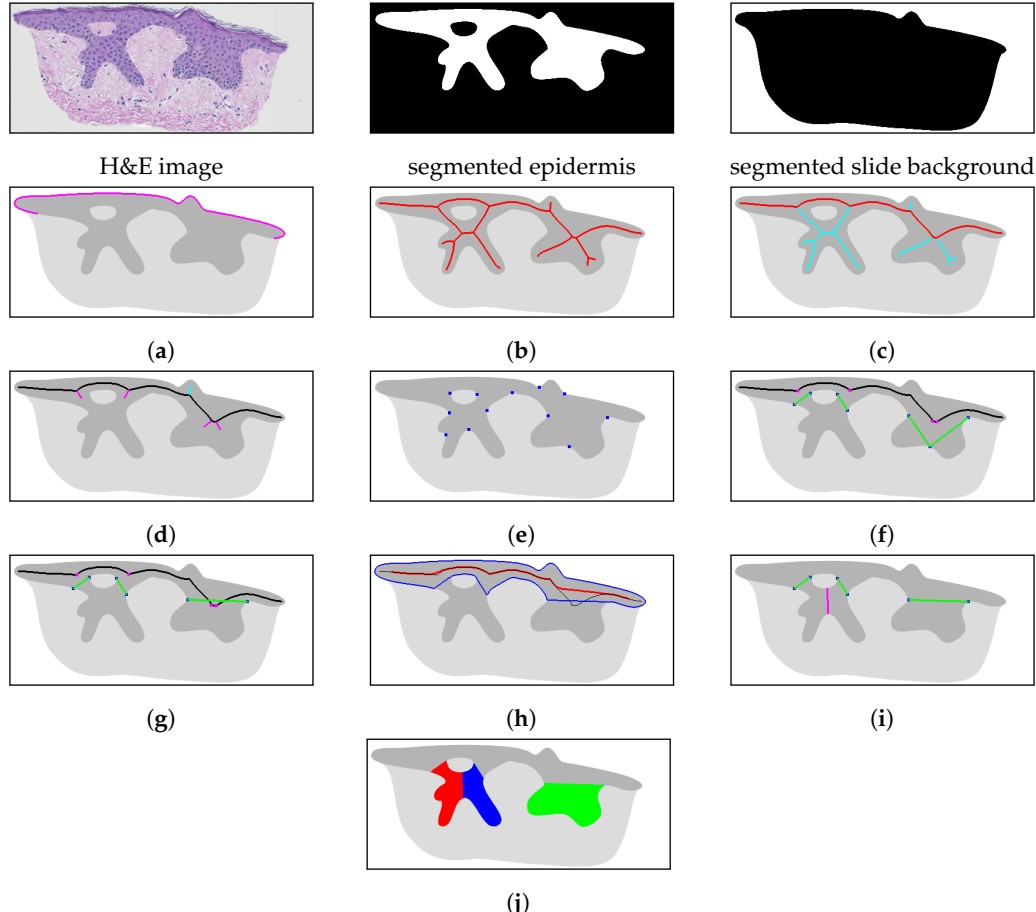

**Figure 7.** The steps of the proposed algorithm: (**a**) determine the epidermis "outer" edge, (**b**) compute the epidermis skeleton, (**c**) determine the initial epidermis main axis (marked red), (**d**) determine the position and orientation of projections ("left side" and "right side" projections are marked cyan and magenta, respectively), (**e**) determine border nodes, (**f**) match border nodes with retes (rete bases and rete roots are marked green and magenta, respectively), (**g**) merge adjacent retes, (**h**) retouch the main axis (the outline of the epidermis base is marked blue, the initial main and retouched main axis are marked black and red, respectively), (**i**) split partially merged retes (the split is marked magenta), and (**j**) compute the width, length and height of each individual rete (here marked with different colors for readability).

## 2.1. Specimen Segmentation

Our approach to specimen segmentation uses color thresholding in the CIELAB color space. CIELAB was designed to be perceptually uniform with respect to human color vision (i.e., the same amount of numerical change in color component values corresponds to about the same amount of visually perceived change). Let us define distance metrics based on CIELAB 1976:

$$\Delta E^*_{Lab}(c_1, c_2) = \sqrt{(L_2^* - L_1^*)^2 + (a_2^* - a_1^*)^2 + (b_2^* - b_1^*)^2} \tag{1}$$

$$\Delta E^*_{ab}(c_1, c_2) = \sqrt{(a_2^* - a_1^*)^2 + (b_2^* - b_1^*)^2} \tag{2}$$

$$C_p^* = \sqrt{(a_p^*)^2 + (b_p^*)^2} \tag{3}$$

where $c_1 = (L_1^*, a_1^*, b_1^*)$ and $c_2 = (L_2^*, a_2^*, b_2^*)$ are two colors in the CIELAB color space, and $C_p^*$ is the chroma (relative saturation) of a color $c_p$.

To identify the "average" background color $c_{Bg}$, we construct a 3D histogram of L*a*b* values of nearly achromatic image pixels, i.e., pixels satisfying the condition $C_p^* < 2\,\text{JND}$ (we partition the

L*a*b* color space into $200 \times 256 \times 256$ bins), and then choose the coordinates of the largest bin. The JND $\approx 2.3$ stands for a "just noticeable difference", as proposed by Mahy et al. [38].

The background mask $M_{\mathrm{Bg}}$ is obtained using hysteresis thresholding (conditions are applied pixel-wise for each image pixel $p$) in the CIELAB color space:

$$\mathrm{IsSimilarToBg}(c_p, \tau) = (\Delta E^*_{Lab}(p, c_{\mathrm{Bg}}) < \tau) \vee (\Delta E^*_{ab}(c_p, c_{\mathrm{Bg}}) < \tau \wedge L^*_p > L^*_{\mathrm{bg}}) \tag{4}$$

$$M_{\mathrm{Bg}} = \mathrm{IsSimilarToBg}(c_p, 2\,\mathrm{JND}) \rightarrow \mathrm{IsSimilarToBg}(c_p, 4\,\mathrm{JND}) \tag{5}$$

where $c_p$ is the color of a pixel $p$, $\tau$ is a color distance threshold, and $\rightarrow$ is a morphological reconstruction. The condition $\mathrm{IsSimilarToBg}(c_p, \tau)$ is satisfied by pixels with color similar to the "average" slide background color $c_{\mathrm{Bg}}$ as well as by pixels brighter than $c_{\mathrm{Bg}}$ but having similar chromaticity. The factors of 2 and 4 have been chosen empirically based on the analysis of color distance histograms for background pixels and tissue pixels (see Section 3.3 for details).

## 2.2. Epidermis Segmentation

Although different methods have been proposed for automatic epidermis segmentation (e.g., [19,21,22]), in our study we decided to use manually prepared epidermis masks, as it is intended to be a proof-of-concept for certain methods of measuring epidermis morphometry (not for the whole automatic diagnostic pipeline). Our epidermis segmentation algorithm has been described in [19].

## 2.3. Determining Normals to the Epidermis Border

Determining normals to the epidermis border is a preliminary step towards identifying the epidermis "outer" edge— it is necessary to distinguish between the "outer" direction (the same as the normal direction) and the "inner" one.

To find normals to the epidermis border we firstly trace the border $L_b$ in the clock-wise direction using Moore-Neighbor tracing algorithm modified by Jacob's stopping criteria [39] to get a sequence of border points $P_1, \ldots, P_n$. Then, to determine the normal $\vec{n}_i$ at the $i$th border point $P_i$ we compute the slope of the vector $\overrightarrow{P_{i-d}P_{i+d}}$ using the four-quadrant inverse tangent, and add a $\pi/2$ constant to obtain a vector orthogonal to $\overrightarrow{P_{i-d}P_{i+d}}$ and pointing "outside" the epidermis (Figure 8). Since the border is a closed shape, if $i - d < 0$ we take the point $P_{n+i-d}$ and if $i + d > n$ we take the point $P_{i+d-n}$. To make results more robust to slight local changes in border's curvature, we set the point distance to $d = 4$.

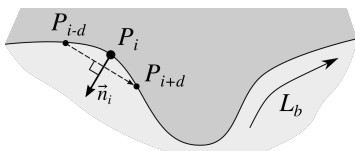

**Figure 8.** Approximating a normal to the epidermis border $\vec{n}_i$ at point $P_i$. The approximated normal is orthogonal to the vector $\overrightarrow{P_{i-d}P_{i+d}}$ (the point distance was set to $d = 4$).

## 2.4. Determining the Epidermis "Outer" Edge

To determine the "outer" edge of the epidermis (the edge adjacent to stratum corneum) for each border point $P_i$ we cast two rays—in the same direction as its normal ($\vec{r}_{\mathrm{n}}$) and in the opposite direction ($\vec{r}_{\mathrm{o}}$). For each ray we compute the distance along the ray to its first intersection with the slide background, $d_{\mathrm{BgN}}$ and $d_{\mathrm{BgO}}$ (Figure 9a). If the ray does not cross the background, we consider the distance to be infinitely large. Since in the dermis there are empty spaces between collagen fibers, to compute the distance to the background we do not use the "raw" mask, $M_{\mathrm{bg}} = \overline{M_{\mathrm{fg}}}$, but $M_{\mathrm{bg}} \circ \mathrm{SE}_{D15}$ (where $SE_{Dn}$ is a disk-shaped structuring element of radius $n$).

The point $P_i \in L_b$ is initially considered as belonging to the "outer" edge if the following conditions are met: (1) $d_{\mathrm{BgN}} < d_{\mathrm{BgO}}$, and (2) $\vec{r}_{\mathrm{n}}$ does not cross epidermis (Figure 9a). The results of

such a test for all border points form a vector of logical values, $M_{\text{Out0}}$, which might be interpreted as a 1D binary mask. We will denote the results of the test "$\vec{r}_n$ vector crosses the epidermis" as $M_{\text{EpX}}$.

Since the initial criteria fail to correctly classify "pockets" in epidermis (Figure 9b), we refine the $M_{\text{Out0}}$ mask in the following three steps. Firstly, for each sequence of 0s in $M_{\text{EpX}}$ (from $i$th to $j$th element) we check if elements $(i-1)$th and $(j+1)$th in $M_{\text{Out0}}$ are both 1s; if it is the case, we consider such a sequence as a "pocket", which should be part of the valid "outer" edge (Figure 9c). Then, to fill tiny gaps between parts of the valid "outer" edge and to suppress artifacts, we perform the following sequence of morphological closings and openings:

$$M_{\text{Out}} = \left( \left( \left( M'_{\text{Out0}} \bullet \text{SE}_{L50} \right) \circ \text{SE}_{L500} \right) \bullet \text{SE}_{L1500} \right) \circ \text{SE}_{L2000} \tag{6}$$

where $\text{SE}_{Ln}$ is a line-shaped structuring element of length $n$. Finally, we retain only the largest segment (Figure 9d).

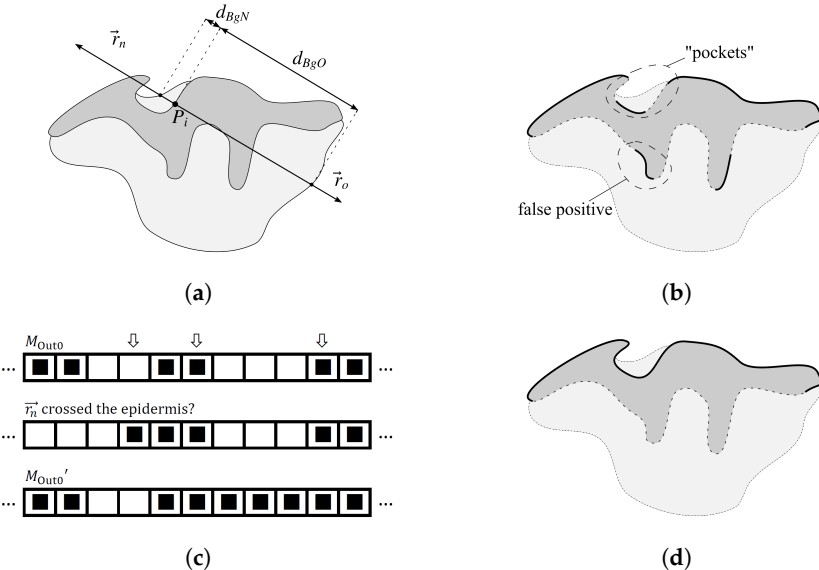

**Figure 9.** Determining the epidermis "outer" edge (the epidermis is shown in dark gray, whereas the rest of the tissue is shown in light gray): (**a**) casting rays $\vec{r}_n$ and $\vec{r}_o$ for each point $P_i$ and measuring the distance to the background for each of those rays, (**b**) the initially detected "outer" edge $M_{\text{Out0}}$ (thick solid line) contains false positives, whereas some sections of the actual "outer" edge (e.g., "pockets") are not properly detected, (**c**) treating all such sequences of 0s in $M_{\text{Out0}}$ that for all their elements the $\vec{r}_n$ vector crosses the epidermis and which are flanked with two sequences of 1s in $M_{\text{Out0}}$ as parts of the valid "outer" edge yields the $M'_{\text{Out0}}$ mask which has issues with most "pockets" fixed, and (**d**) performing the sequence of morphological closings and openings on the $M'_{\text{Out0}}$ mask fixes the remaining issues with "pockets" and false positives—the final version of the epidermis "outer" edge, $M_{\text{Out}}$, is marked with a thick solid line.

## 2.5. Computing the Epidermis Skeleton

To compute the skeleton of the epidermis region, we firstly skeletonize it using an augmented fast marching method proposed by Telea and van Wijk [40] with pruning threshold $t = 100$ and then additionally perform the medial axis transform on the resulting mask to remove excess pixels near junctions.

The pruning threshold $t$ has a precise geometrical meaning: all skeleton branches caused by boundary details shorter than $t$ pixels are pruned [40]. At resolution $0.44\,\mu\text{m/px}$ the threshold value of $t = 100$ corresponds to roughly $30\,\mu\text{m}$, which turned out to be the minimum length of a boundary detail when a pathologist was able to decide whether that detail was actually a rete.

### 2.6. Determining the Epidermis Main Axis

The epidermis main axis is a centerline of the epidermis base, i.e., the epidermal region with retes passed over (Figure 4). We determine the epidermis main axis to identify pixels being candidates for rete roots (as all roots of retes are located on that axis) and to distinguish between projections branching off towards the underlying skin layers and towards the stratum corneum (the outermost skin layer).

A naive solution to determine the epidermis main axis would be to find the longest path in a graph $G$ spanned on the skeleton of the epidermis. However, there are two problems with such an approach: (1) if holes are present in epidermis (e.g., due to partially merged retes) the longest path in $G$ may not necessarily run along the epidermis "outer" edge (Figure 10a), and (2) simply taking the longest path in $G$ would fail if there was a long rete near an edge of the epidermis region (Figure 10b).

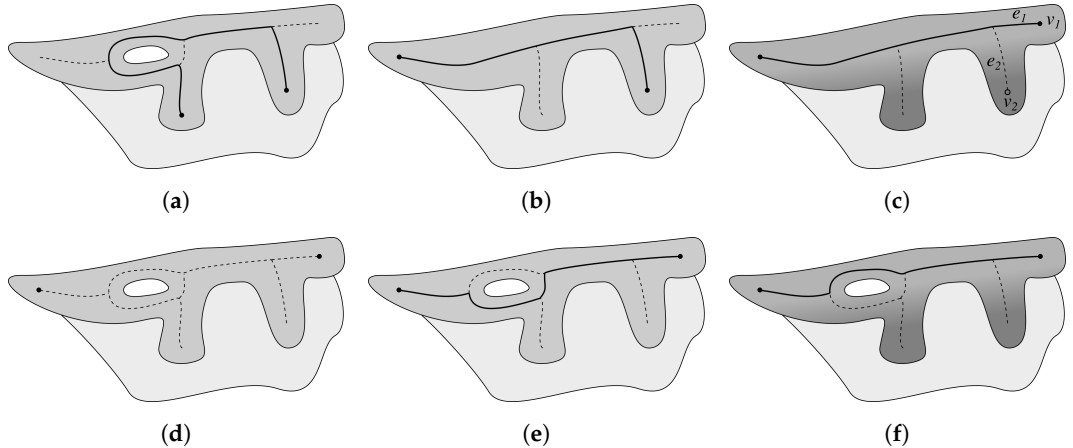

**Figure 10.** Issues when finding the epidermis main axis and their solutions: (**a**) holes in epidermis cause the longest path in $G$ not to run along the epidermis outer edge, (**b**) the problem with holes can be resolved by locating main axis endpoints based on the analysis of the skeleton of a filled epidermis region yet long retes near ends of the epidermis region still cause the longest path in $G$ to omit the last segment of the true main axis, (**c**) the problem with long retes can be resolved by ranking endpoints based on their depth ($|e_1| - D(v_1) > |e_2| - D(v_2)$), (**d**) endpoints determined using the $G_f$ skeleton are also valid in $G$, (**e**) simply taking the longest path between endpoints does not work, and (**f**) finding the shortest path when edges are weighted according to the sum of depth of pixels forming a given edge gives the proper result. In each step the computed main axis is marked as a thick solid line.

To address these issues, we firstly determine the two endpoints of the main axis by choosing the best path in a graph $G_f$ spanned on the skeleton of epidermis region with holes filled (skeletonization is performed using the same procedure as described in Section 2.5), which ensures that $G_f$ is acyclic and that the position of all leaves in $G$ is retained in $G_f$ (Figure 10b). We rank paths in $G_f$ using the score including a penalty based on the depth of the path endpoints:

$$\text{Score}(\pi) = \sum_{e_i \in \pi} w(e_i) - (D(v_0) + D(v_n)) \tag{7}$$

where $\pi = (e_1, \ldots, e_p)$ is a path in a graph $G_f$, $w(e)$ is the weight of the edge $e$ (here: number of pixels forming this edge), $D(v)$ is the geodesic distance of $v$ from the epidermis "outer" edge, and $v_0$ and $v_n$ are endpoints of $\pi$. We consider all possible paths from all leaves in $G_f$ and choose endpoints of the path with the highest score as the main axis endpoints $v_1^e$ and $v_2^e$ (Figure 10c). To determine the actual route of the main axis, we use Dijkstra's algorithm to find the longest path between $v_1^e$ and $v_2^e$ in $G$ [41]. Edges are weighted according to the mean depth of pixels $p_i$ forming a given edge:

$$w(e) = \sum_{p_i \in e} D(p_i). \tag{8}$$

It ensures that we choose a path running through the epidermis base.

### 2.7. Determining the Position and Orientation of Retes

To determine the location and the orientation of retes we firstly compute these parameters for all projections branching off from the main axis and then pick only projections corresponding to retes.

For each edge $e_i = (v_q, v_r)$ where $v_q$ lies on the main axis we choose $v_q$ as the projection root. To get the approximated orientation of the projection $\varphi_p$, we choose the pixel $v_p$ (the end of a rete "stump") which is $L_p = \min(|e_i|, l_{max})$ pixels away from $v_q$ along $e_i$ and compute the four-quadrant inverse tangent of the $\overrightarrow{v_q v_p}$ vector (Figure 11). We set $l_{max} = 10$.

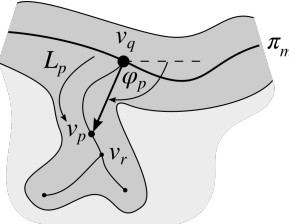

**Figure 11.** The schema of determining the orientation of a rete ($\varphi_p$) based on the location of a rete root ($v_q$) and the corresponding end of a rete stump ($v_p$).

We then assign each projection into one of two groups—"left" projections and "right" projections—based on the side of the main axis it sticks out. To perform such an assignment, we dilate the main axis using a disk-shaped structuring element of radius $l_{max}$, split it in half using the main axis and two rays (the one cast from the first node of the axis in the direction opposite to the direction of the first axis segment, and the other cast from the last node in the same direction as the last segment), and check which ends of rete "stumps" overlap with which half (Figure 12).

For each projection in each group we cast a ray in the same direction as the orientation of a given projection and measure the distance along that ray to its first intersection with the slide background. Projections in the group for which the mean distance to the slide background is larger are considered retes.

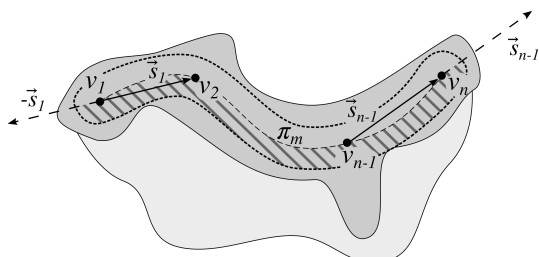

**Figure 12.** Distinguishing between "left" and "right" projections. The dilated main axis $\pi_m$ region is split in halves by the main axis $\pi_m$ combined with rays $-\vec{s}_1$ (cast from $v_1$) and $\vec{s}_{n-1}$ (cast from $v_n$). Rete stumps in the striped half are considered "right" projections and in the other half—"left" projections.

### 2.8. Determining Border Nodes

To choose endpoints of rete bases we firstly determine candidate points on the epidermis border. A border point is considered a candidate if it is in the center of a border protrusion. We cannot simply take points on the border closest to the rete root as for broad and long retes the main axis determined using the skeleton is heavily deformed—in such case we would choose points in the middle of rete's length instead of points near its base (Figure 13).

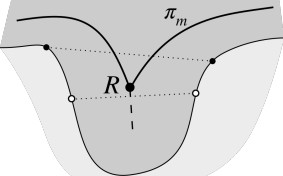

**Figure 13.** To choose endpoints of rete bases we firstly determine candidate points, located in centers of border protrusions (black dots). Simply taking the points on the border closest to the rete root *R* (white dots) yields incorrect results for broad and long retes as in such case the main axis $\pi_m$ determined using the skeleton is heavily deformed (the junction point *R* is shifted towards the center of a rete).

We compute relative changes in the border slope between consecutive border points (see Section 2.3) and represent it as a function of border run length (Figure 14a). The relative orientation of the vector $v_1$ with respect to the vector $v_2$ is given by:

$$\Delta\varphi(v_1, v_2) = \begin{cases} \varphi_1 - \varphi_2 & \text{if } \varphi_1 - \varphi_2 \in [-\pi, \pi] \\ \varphi_1 - \varphi_2 + 2\pi & \text{if } \varphi_1 - \varphi_2 < -\pi \\ \varphi_1 - \varphi_2 - 2\pi & \text{if } \varphi_1 - \varphi_2 > \pi \end{cases} \tag{9}$$

where $\varphi_1$ and $\varphi_2$ are the absolute orientations of vectors $v_1$ and $v_2$, respectively. Note that $\Delta\varphi \in [-\pi, \pi]$.

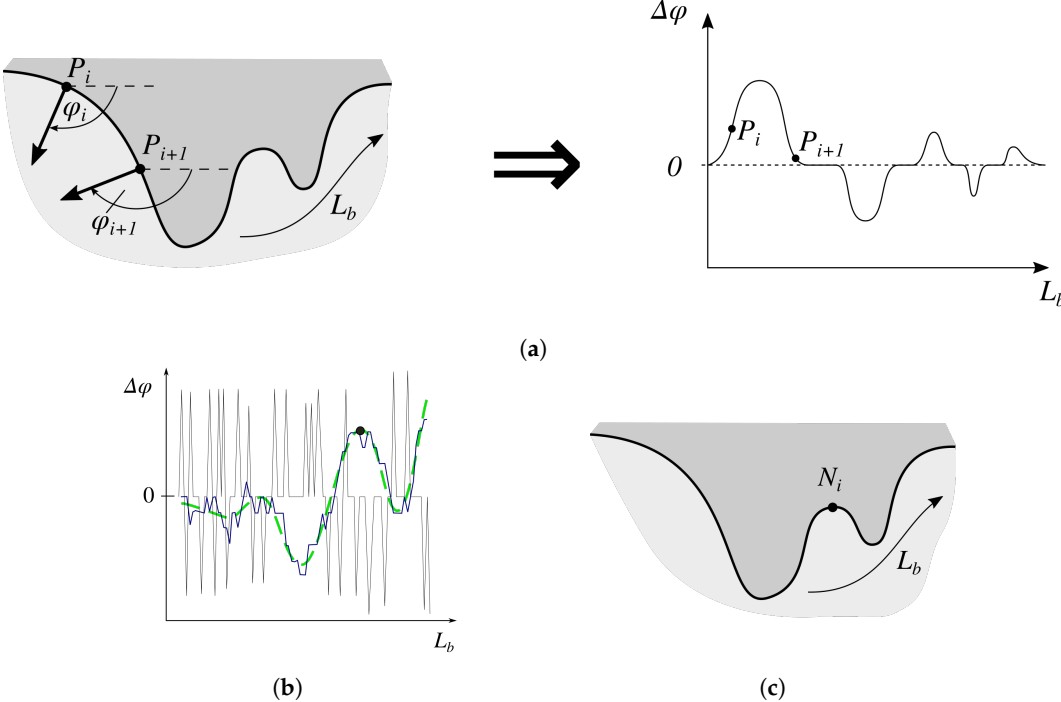

**Figure 14.** Determining border nodes: (**a**) represent relative changes in the border slope between consecutive border points as a function of border run length, (**b**) filter the function of relative changes in the border slope between consecutive border points w.r.t. the border run length (thin solid line—the original signal $f$; thick solid line—$f$ averaged using the simple moving average filtering, $f_{\text{avg}}$; dashed line—$f_{\text{avg}}$ after applying a low-pass filter), (**c**) the identified node.

To remove tiny fluctuations we firstly use the simple moving average filtering with window size $w_s = 15$. To get the general trend, we then perform low-pass filtering using a minimum-order low-pass finite impulse response (FIR) filter with normalized passband frequency 0.1 rad/px, stopband frequency 0.15 rad/px, passband ripple 0.01 dB, and stopband attenuation 65 dB, designed using a Kaiser window method (for boundaries consisting of less then 100 pixels we use stopband attenuation

50 dB). In both cases, since the signal corresponds to a closed curve, when smoothing we wrap the signal to prevent artifacts on its ends. Figure 14b shows the results of the compound filtering.

We find locations of such local maxima (peaks) which height $h$ and prominency $p$ is sufficiently large, as higher peaks correspond to sharper turns and more prominent peaks represent a more constant trend. We chose $h > 0.025$ and $p > 0.0015$, respectively (Figure 14c). Again, we analyze a wrapped signal so that peaks on the edge of a non-wrapped signal are correctly detected.

### 2.9. Matching Border Nodes with Retes

For each rete root $R$ we find its right and left base node, corresponding to endpoints of the rete base edge. A candidate for any of such nodes, $N_i$, must lie close to the rete root (we set $|RN_i| < 300$ as a reasonable limit) and must lie on the epidermis "inner" edge.

Candidate nodes are then assigned to the "left" and "right" group as follows: if $\Delta\varphi(\vec{n}_i, \vec{p}) > 0$ then the node is considered a "left" candidate, otherwise it is considered a "right" candidate ($\vec{p}$ is a rete vector and $\vec{n}_i = \overrightarrow{RN_i}$ is a vector from the rete root $R$ to a node $N_i$). Within each group we firstly try select closest node for which $|\Delta\varphi(\vec{n}_i, \vec{p})| \in (10°, 150°)$. If no such node is found, we relax the criterion and choose from all the nodes in the group.

Occasionally, due to the lack of concavities in epidermis border near a rete root, one of the following scenarios might happen for one side of a rete: (1) a node too far away from the rete root is selected, (2) a node on the incorrect side of the main axis is selected, or (3) no node is selected at all. Figure 15 shows all these circumstances. In such cases we choose the border point located on the ray cast from the rete root in the direction determined as follows: we take the direction from rete root to the "proper" node and mirror it w.r.t. the rete vector (Figure 16). The "proper" node is the node located closer to the rete root if two nodes were initially found.

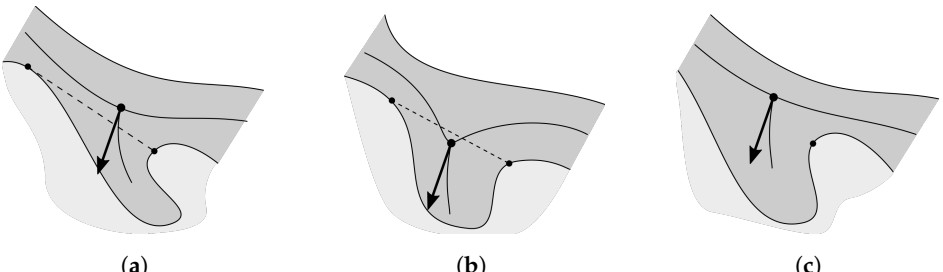

|     (a)     |     (b)     |     (c)     |

**Figure 15.** Problematic cases when determining rete's nodes: (**a**) a node too far away from the rete root is selected, (**b**) a node on the incorrect side of the main axis is selected, and (**c**) no node is selected at all.

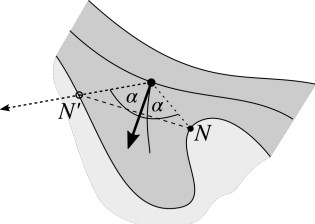

**Figure 16.** The "proper" node $N$ is mirrored with respect to the rete vector to obtain a new node $N'$.

### 2.10. Merging Adjacent Retes

Retes for which the distance between their roots is less than 50 px (approx. 20 µm) are treated as a group. Within each group we choose two most distant roots ($R_a$ and $R_b$) and use vectors of their corresponding retes to determine the mean vector of the joint rete $\vec{p}_{avg}$. To determine base nodes of the joint rete, for each node $N_i$ forming the base of a rete belonging to the group, we compute the relative orientation of the vector $\vec{n}_i = \overrightarrow{ON_i}$ (where $O$ is equidistant from $R_a$ and $R_b$) with respect to the mean vector $\vec{p}_{avg}$ according to (9), and then choose one node with largest and one node with smallest relative orientation $\alpha = \Delta\varphi(\vec{n}_i, \vec{p}_{avg})$. To determine the position of the new rete root, we find a point $R'$ on the

main axis closest to the center of mass of joint roots $R_{avg}$ (we take into account rete roots of all retes belonging to the group). Figure 17 shows the procedure of merging adjacent retes.

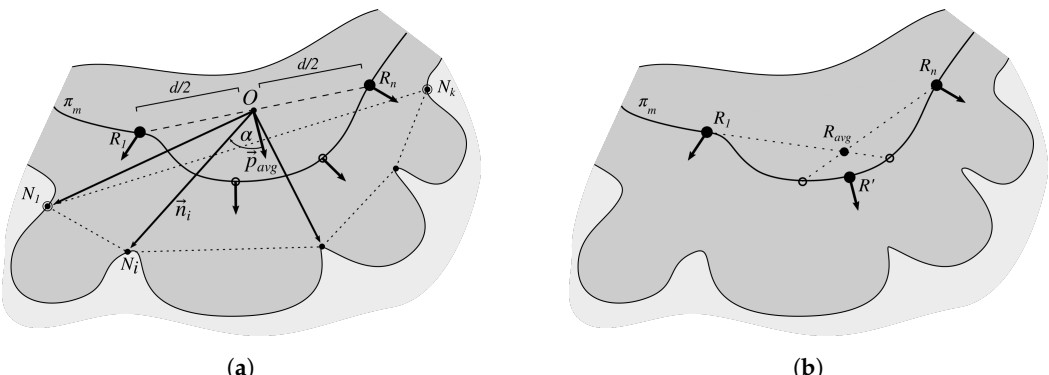

|  |  |
|:--:|:--:|
| (**a**) | (**b**) |

**Figure 17.** Merging the adjacent retes: (**a**) the two most distant retes roots are $R_1$ and $R_n$ and the base nodes of the new (joint) rete are $N_1$ and $N_k$, (**b**) the root of the new (joint) rete $R'$ is the point on the main axis located closest to the center of mass of joint roots $R_{avg}$.

## 2.11. Retouching the Main Axis

After rete bases are determined, we retouch the main axis so that it runs only through the epidermis base (and not through upper parts of retes). To do this, we firstly determine the epidermis base mask $M_{EpBase}$ by reconstructing the original main axis into an epidermis mask with rete bases removed (rete bases are rendered using Bresenham's line algorithm [42]). Then, we repeat the procedure of finding the "shallowest" path described in Section 2.6 using a skeleton of $M_{EpBase}$ (obtained by applying the procedure described in Section 2.5, this time with threshold $t = 200$ to retain only skeleton sections forming the main axis) and the same endpoints as in the original main axis.

## 2.12. Splitting Partially Merged Retes

Some retes in the specimen are partly joined with their adjacent retes. To be able to measure the height of each individual rete, it is necessary to determine the approximate delimitation of such joined retes (Figure 18).

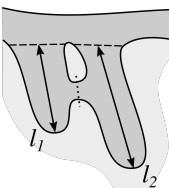

**Figure 18.** Joined retes must be delimited (the dotted line) before measuring their morphometry, otherwise both would have for example the same length equals to $l_2$.

We start delimiting retes with tracing boundaries of each hole in epidermis and then repeat the following procedure until all boundaries are processed. The procedure analyzes holes row by row (Figure 19) and in the $i$th pass we consider only holes forming the $i$th row.

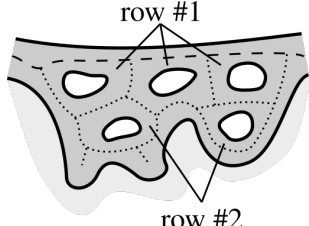

**Figure 19.** Rows of holes. The first row is formed by such cycles around holes that in each cycle at least one edge belongs to the main axis (a dashed line). The second row is formed by cycles adjacent to the first row etc.

In the $j$th iteration we firstly identify nodes in the original skeleton forming the cycle around the $j$th hole, which we will denote as $V_j$. Therefore we firstly reconstruct a mask containing the hole border into the $(M_{\text{Ep}} \setminus M_{\text{Skel0}})$ mask, followed by its dilation with a square-shaped structuring element of size $3 \times 3$ (so that it includes skeleton nodes), and performing the logical AND operation with skeleton nodes.

If its the first pass, we skip all holes for which their boundaries do not overlap with the main axis. For the rest of holes we identify two edges ("trunks" of two retes) flanking a given hole and take the mean of their orientations as the orientation of the splitting ray. The splitting ray is cast from the point on the hole border which is most distant from the epidermis outer edge (in Euclidean metric), and it splits the epidermis until it reaches the background region. We trace the number of times a given skeleton node has been processed.

In the subsequent iterations ($i > 1$), for each cycle we firstly find those of its nodes which were already processed, but only once. Let us denote this set as $V_{j1}$. We then find such two edges, which one endpoint belongs to $V_j$ and the other to $V_{j1}$, compute the approximated orientation of those edges using the endpoints of their 10 px-long stumps rooted at the common nodes (the procedure is analogous to the one described in Section 2.7), and use the mean of their orientations as the orientation of the splitting ray (the procedure for choosing the casting point is the same as described above).

Figure 20 shows the splitting procedure applied to holes from the first and the second row. The binary mask holding a superposition of all splitting rays, $M_{\text{Split}}$, is used when computing base width and height of individual retes.

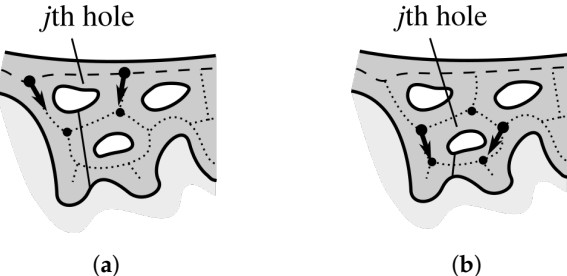

(**a**)　　　　　　　　(**b**)

**Figure 20.** Splitting partially merged retes: (**a**) a hole in the first row, (**b**) a hole in the second row. Nodes $V_j$ are marked with circles, orientation of edges flanking the hole is marked with an arrow, and the splitting ray is marked with a solid line.

### 2.13. Computing the Morphometry of Retes

In order to measure the morphometry (width, length and height) of individual retes, we firstly compute a mask $M_{\text{Proj}}$ containing all rete regions:

$$M_{\text{Proj}} = M_{\text{Ep}} \setminus (M_{\text{Base}} \cup M_{\text{Split}}).$$

Then an individual rete is extracted by reconstructing its edge into $M_{\text{Proj}}$.

The base width of a rete corresponds with the length of its base (i.e., the Euclidean distance between base endpoints). To compute the length of a rete, we compute the geodesic distance transform of the projection's region with seeds located at the projection's base and take the maximum value found in the transformed image (Figure 21a). To compute the height of a rete, for each boundary pixel $P$ we measure its distance from the line $k : A\,x + B\,y + C = 0$ containing the projection's base and take the maximum value across all boundary points (Figure 21b). Assuming $N_1 = (x_1, y_1)$ and $N_2 = (x_2, y_2)$ are endpoints of the projection's base, the straight line coefficients are given by:

$$A = \begin{cases} 1 & \text{if } x_1 = x_2 \\ \frac{y_1-y_2}{x_1-x_2} & \text{otherwise} \end{cases} \quad B = \begin{cases} 0 & \text{if } x_1 = x_2 \\ -1 & \text{otherwise} \end{cases} \quad C = \begin{cases} -x & \text{if } x_1 = x_2 \\ y_1 - \frac{y_1-y_2}{x_1-x_2}x_1 & \text{otherwise} \end{cases} \quad (10)$$

and the distance from the point $P = (x_P, y_P)$ to the line $k$ is:

$$d(P,k) = \frac{|A\,x_P + B\,y_P + C|}{\sqrt{A^2 + B^2}}. \quad (11)$$

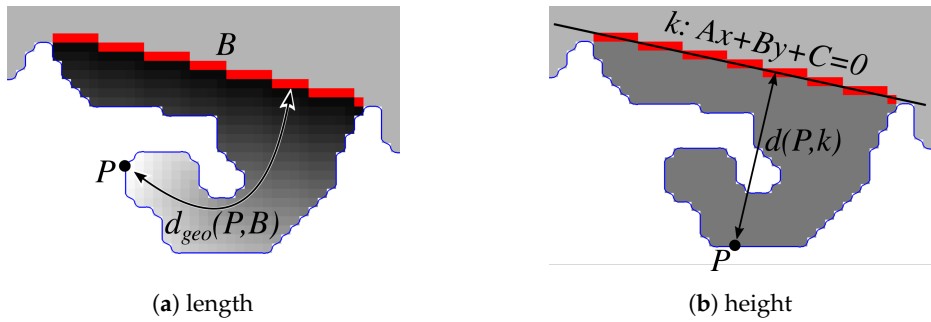

(**a**) length          (**b**) height

**Figure 21.** Determining the projection's morphometry: (**a**) the projection's length is approximately the maximum value of the geodesic distance transform of the projection's region with seeds located at the projection's base, (**b**) the projection's height is the maximum distance of a boundary point $P$ from the straight line $k : Ax + By + C = 0$ containing the projection's base.

## 3. Results

### 3.1. Image Dataset

The presented method has been evaluated on 25 skin whole slide images (WSIs), obtained from Jagiellonian University Medical College. The set included cases of lentigo (9), dysplastic nevus (11), and melanoma (5). Images were captured under $10\times$ magnification ($0.44\,\mu\text{m}/\text{px}$) on Axio Scan.Z1 slide scanner and saved into TIFF format.

For each image the ground truth consisting of the epidermis segmentation (a binary mask) as well as information about each individual rete (the approximated location of its base endpoints and its length) was prepared manually by an experienced dermatopathologist. The manual epidermis segmentations were prepared using GIMP image processing program and the information about individual retes were gathered using the "ROI Manager" functionality of ImageJ image processing program.

### 3.2. Evaluation Metrics

To evaluate our method we used both quantitative and qualitative measures. The aim of the quantitative analysis was to answer the two following questions: (1) what was the proportion of retes detected in the correct location?, and (2) how severe is the problem of overdetection (i.e., false positives)? The qualitative analysis helps to assess how closely the morphometry of automatically segmented retes match the morphometry of manually segmented ones.

Both manually and automatically segmented retes bases are actually segments in the same 2D image space. Since it is virtually impossible for any pair of manually and automatically segmented retes to perfectly overlap, as typically the corresponding endpoints would be located at least a few pixels away, we defined the following matching criterion: in order to consider a rete $R_1$ as matching a rete $R_2$ the distance between the centers of their bases must be at most $l_2/2$, where $l_2$ is the length of $R_2$'s base.

Let us define the set of automatically segmented retes as $\mathbf{R}_A$ and the set of manually segmented retes as $\mathbf{R}_M$. If either: (1) $R_A^i \in \mathbf{R}_A$ is matched by exactly one $R_M^j \in \mathbf{R}_M$ and $R_M^j$ is matched only by $R_A^i$, or (2) $R_A^i \in \mathbf{R}_A$ is matched by exactly one $R_M^j \in \mathbf{R}_M$ and $R_M^j$ is not matched with any rete from $\mathbf{R}_A$, or (3) $R_M^j \in \mathbf{R}_M$ is matched by exactly one $R_A^i \in \mathbf{R}_A$ and $R_A^i$ is not matched with any rete from $\mathbf{R}_M$, then we say that these two retes are matched exclusively. If a given manually segmented rete is matched by multiple automatically segmented retes, then those automatically segmented retes should be merged. On the other hand, if a given automatically segmented rete is matched by multiple manually segmented retes, then that automatically segmented rete should be split.

To measure the performance of our method we used the Jaccard index. The Jaccard index is a statistic used for comparing the similarity and diversity of finite sample sets, and is defined as the size of the intersection divided by the size of the union of the sample sets:

$$J(A,B) = \begin{cases} \frac{|A \cap B|}{|A \cup B|} = \frac{|A \cap B|}{|A|+|B|-|A \cap B|} & \text{if } A \cup B \neq \varnothing \\ 1 & \text{otherwise} \end{cases} \tag{12}$$

In our case the two sets are a set of manually segmented retes—the ground truth (denoted as MAN)—and a set of automatically segmented retes (AUT).

We computed two metrics based on the Jaccard index. In the restrictive one, the intersection set consists only of retes matching exclusively:

$$J_{\text{strict}} = \frac{\#\text{ true positive MANs}}{\#\text{ MANs} + \#\text{ false positive AUTs}} \tag{13}$$

In the relaxed one, non-exclusively matching automatically segmented retes (i.e., both retes to be merged and to be split) also belong to the intersection set:

$$J_{\text{relaxed}} = \frac{\#\text{ MANs} - \#\text{ false negative MANs}}{\#\text{ MANs} + \#\text{ false positive AUTs}} \tag{14}$$

We included the relaxed version as in many cases the decision made by a pathologists whether to treat a projection in epidermis as one rete with two tips or two separate retes is arbitrary (Figure 22). In a perfect case, when all ground truth retes are matched and there are no automatically segmented retes without a match, the Jaccard index equals 1. Otherwise its value is in the range $[0, 1]$.

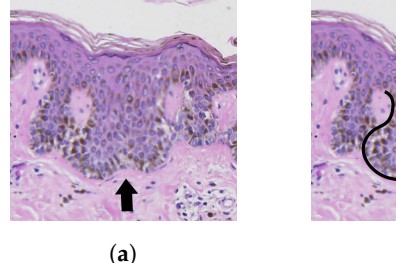 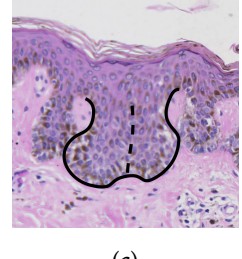

(**a**) (**b**) (**c**)

**Figure 22.** The splitting dilemma: the central region in (**a**) can be treated either as a single rete with two tips as in (**b**) or as two merged retes as in (**c**).

Before computing Jaccard indexes we fine-tune the sets of manually and automatically segmented retes. We reject all retes which roots are located at most $\tau_{\text{Lep}} = 250\,\text{px}$ from main axis endpoints, as edges of a skin specimen are often deformed due to mechanical slicing. Similarly, we reject all retes which fail to meet the following morphometric criteria: (1) their width $w$ is too small ($w < w_{\min}$), and (2) their length $l$ is too small ($l < l_{\min}$). We set $w_{\min} = 25\,\text{px}$ and $l_{\min} = 50\,\text{px}$, as for smaller projections the decision whether to consider it as a rete or just a slicing artifact is arbitrary (even by experienced pathologists). For automatically segmented retes we also reject retes for which their length-to-width ratio is less then 0.5. However, all those three criteria are not applied to automatically segmented retes which are adjacent to holes in epidermis base (as such holes are always signs of bridging between retes).

For retes matching exclusively we calculated standard statistical measures describing the distribution of relative errors for both lengths of rete bases and rete lengths. For the true value of a quantity, $x$, and the inferred value, $\hat{x}$, the relative error is defined by

$$\delta x = \frac{\hat{x} - x}{x}. \tag{15}$$

### 3.3. Parameter Selection

In our specimen segmentation method we have chosen both the chroma threshold for histogram construction and factors in (5) empirically, based on the analysis of color distance histograms for background pixels and tissue pixels (Figure 23). We evaluated that method on large extracts from 12 skin WSIs from three different laboratories and captured using four different devices (overview in Table 1), to cover most of the variations which one would encounter in regular clinical practice. The mean true positive rate (TPR) and the mean intersection-to-union area ratio (i.e., Jaccard index) of the calculated mask to the ground-truth mask across all images were 0.999 and 0.986, respectively (prior to computing these statistics all calculated and ground truth masks were opened with a disk-shaped structuring element of radius 15, as it is a pre-processing operation we use when determining the epidermis "outer" edge).

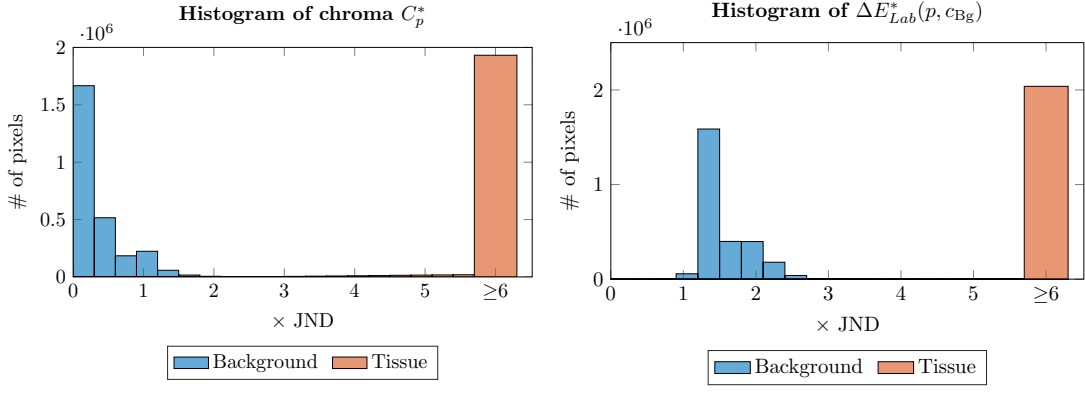

**Figure 23.** Typical histograms of chroma $C_p^*$ and distance to the "average" background color $\Delta E_{Lab}^*(p, c_{\text{Bg}})$ for the same WSI. Note that in both cases for most background pixels $C_p^* < 2\,\text{JND}$ and $\Delta E_{Lab}^*(p, c_{\text{Bg}}) < 2\,\text{JND}$.

**Table 1.** WSI datasets used to evaluate the specimen segmentation method. Each dataset consisted of 3 images.

| Set | Mag. | μm/px | Image Format | Device | Laboratory |
|-----|------|-------|--------------|--------|------------|
| 1 | 20× | 0.345 | TIFF | Olympus BX51+ Pike F505C VC50 | Jagiellonian Univ. MC |
| 2 | 10× | 0.44 | TIFF | Axio Scan.Z1 | Jagiellonian Univ. MC |
| 3 | 40× | 0.25 | JPEG2000 q70 | Aperio AT2 | Univ. of Michigan [43] |
| 4 | 40× | 0.25 | JPEG q30 | Aperio ScanScope CS$^2$ | Univ. of British Columbia [44] |

In the epidermis skeleton computation step, the pruning threshold $t$ has a precise geometrical meaning: all skeleton branches caused by boundary details shorter than $t$ pixels are pruned [40]. In practice, a pathologist is able to decide whether a boundary detail could be a rete only if the detail has the length of at least two average-sized keratinocytes (keratinocytes are cells forming the cellular epidermis). Since the minimum epidermal plate thickness found in human body (in forearm dorsal) varies between 45–65 μm [45,46] and the diameter of an average keratinocyte is 12–16 μm [47], the distance from the epidermis main axis to the tip of a boundary detail should be at least about 50 μm (at resolution 0.44 μm/px it corresponds to 114 px). However, in order not to miss potential retes at extremely thin epidermal sections (e.g., in melanoma), we adopted the pruning threshold value $t = 100$ (Figure 24).

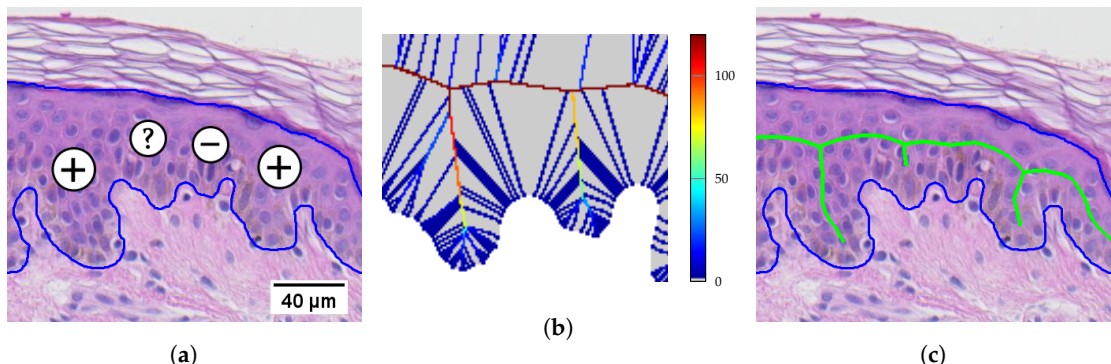

(**a**)  (**b**)  (**c**)

**Figure 24.** A section of epidermis: (**a**) two boundary details are definitely retes ("+"), one might be a rete ("?"), and one is definitely not a rete ("−"); (**b**) the $U$ difference field (see [40]) for the two central boundary details; (**c**) the skeleton obtained by thresholding the $U$ difference field with $t = 100$.

When retouching the main axis, the choice of the pruning threshold $t$ is arbitrary, as it only affects the length of sections of the main axis which are cut off from each end of the axis. On the other hand, larger values of $t$ results in faster computations as there are less graph edges to process when determining the retouched main axis. We chose the threshold value of 200, which at resolution 0.44 μm/px corresponds roughly to 90 μm.

### 3.4. Performance

In total, the dataset consisted of 992 manually segmented retes, whereas our method detected 894 retes—825 were detected correctly (83%), 147 manually segmented retes were not detected, in 20 cases either one manually segmented rete was assigned multiple automatically segmented retes or vice versa (automatically segmented retes should be split or merged), and 42 detected retes were actually false positives.

Figure 25 shows the distribution of retes as well as individual error types with respect to rete's length and width. Most "should split or merge" errors are confined to micro-retes: 62% of all "should split or merge" retes had the width of less than 140 px and the length of less than 90 px whereas only 15% of all retes had such dimensions.

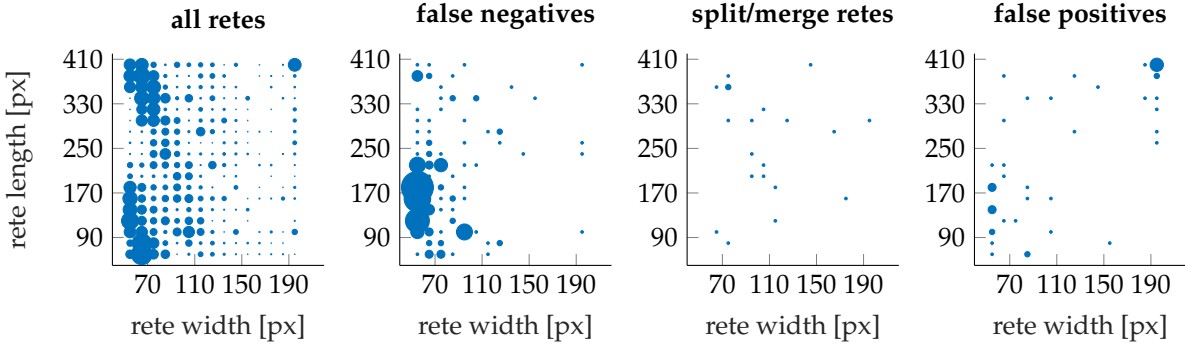

**Figure 25.** Scatter plots of the distribution of retes wrt. to their length and width. The marker scale for all error plots is the same, yet (for clarity) it differs from the marker scale for the "all retes" plot.

Figure 26 shows a histogram of incorrectly detected retes w.r.t. the rete length. It can be noted, that the proportion of misdetections rapidly decreases with the length of a rete and stabilize at 10–17% for retes longer than 90 px (which make 77% of all retes). The highest shares of errors were among retes shorter than 80 px, where up to 65% of retes were not detected. A high number of false positives for retes longer than 200 px (15 cases) is caused by skin appendages (mainly hair follicles) misclassified as retes. Both retes and hairs have similar morphometry, yet their tissue structure differs (Figure 27). However, to the best of our knowledge, no automatic method for the segmentation of skin appendages in histopathological images has been proposed so far.

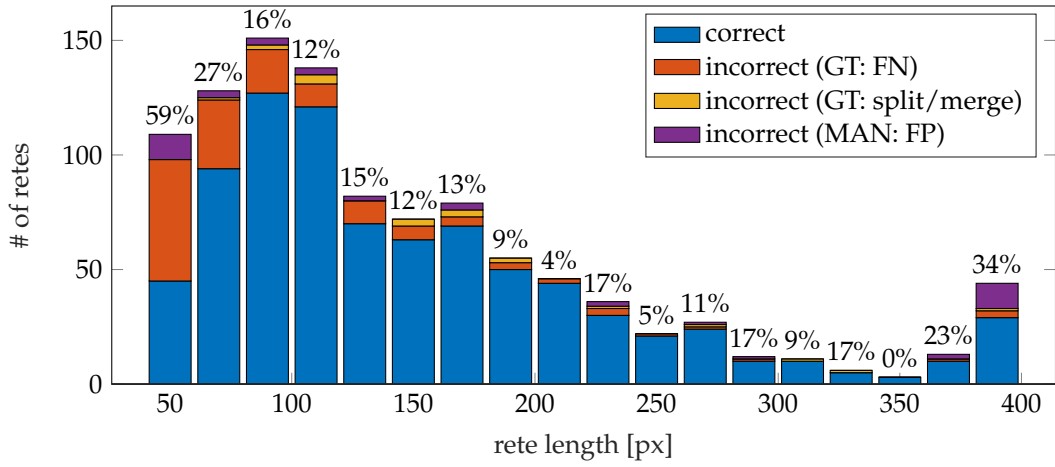

**Figure 26.** A histogram of the share of incorrectly detected retes with respect to the rete length (the first bin also includes all retes shorter than 50 px and the last bin also includes all retes longer than 410 px).

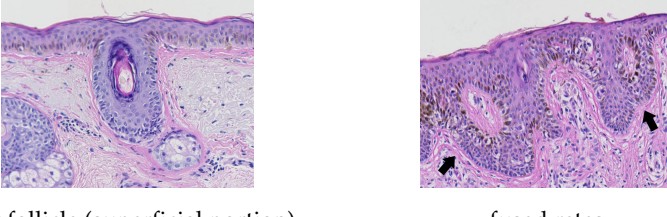

<center>hair follicle (superficial portion)                              fused retes</center>

**Figure 27.** Morphometric similarity between skin appendages (e.g., hair follicles) and fused rete ridges.

Figure 28 shows boxplots of Jaccard indexes computed on the set of $n = 25$ of images. The median and interquartile range (IQR) for $J_{strict}$ were 0.80 and 0.10, whereas for $J_{relaxed}$ they equaled 0.81

and 0.11, respectively. The lower adjacent value was nearly 0.70 for both the strict and the relaxed version. The Jaccard indexes computed for the aggregated data (from all slides) were $J_{strict} = 0.798$ and $J_{relaxed} = 0.817$.

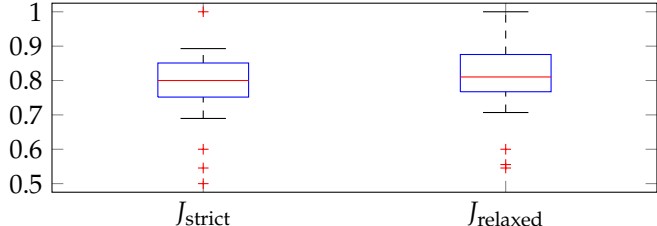

**Figure 28.** Boxplots of Jaccard indexes computed on the set of $n = 25$ of images.

Figure 29a,b shows histograms of relative errors for estimating rete width and length, respectively. The mean absolute value of relative errors ($|\delta x|$) was 20% for the width and 16% for the length. Both distributions of relative errors were right-skewed, with skewness equals to 0.79 for width errors (moderate skewness) and 3.42 for length errors (high skewness), which means that our method tends to overestimate both width and length of retes rather than to underestimate them. There is no point in making a similar analysis for rete height, as even a small change in the orientation of rete's base has a significant impact on its height estimate.

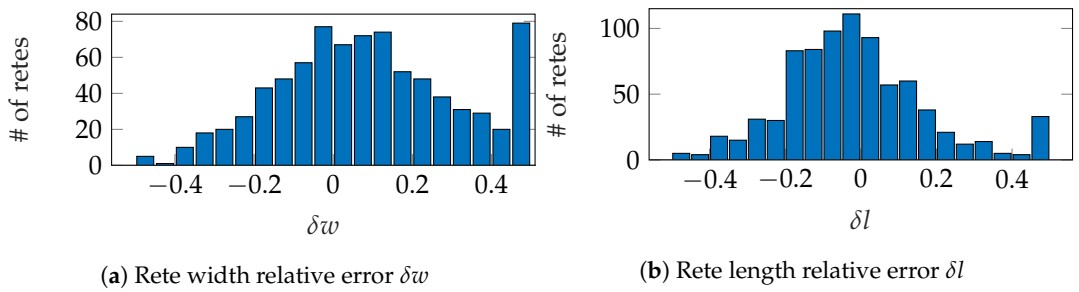

(**a**) Rete width relative error $\delta w$                    (**b**) Rete length relative error $\delta l$

**Figure 29.** Histograms of relative errors for: (**a**) rete width, (**b**) rete length. The first bin also includes errors smaller than $-0.50$ and the last bin also includes errors larger than 0.50.

Our method has three main limitations (see Figure 30):

1. the algorithm incorrectly takes long retes at the end of the lesion as part of the main axis (since usually such a rete is close to the lesion boundary, it is difficult to distinguish a short "tip" of the epidermal plate from a rete);
2. long retes which do not grow approximately perpendicularly into dermis, but rather are joined with the epidermal plate in multiple places along their boundary, will usually be split by the algorithm as if they were series of short retes fused together;
3. our method fails to identify the rete base for a group retes fused near the epidermal plate if total width of such a group is too large.

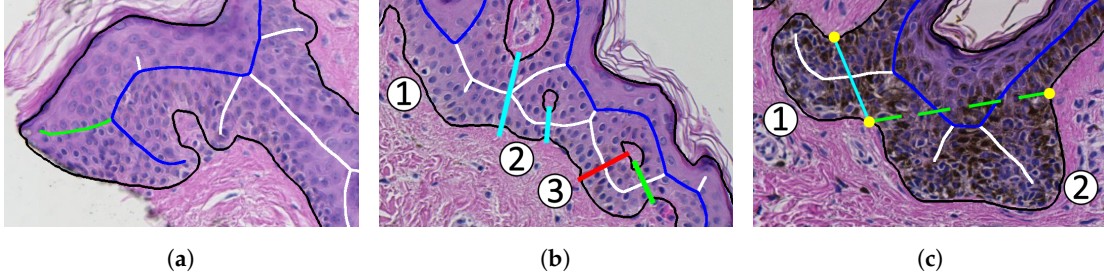

|       |       |       |
|:-----:|:-----:|:-----:|
| (**a**) | (**b**) | (**c**) |

**Figure 30.** The three main limitations of the proposed method: (**a**) the algorithm incorrectly takes long retes at the end of the lesion as part of the main axis (the main axis is marked blue, whereas the proper ending of the main axis is marked green); (**b**) long retes which do not grow approximately perpendicularly into dermis, but rather are joined with the epidermal plate in multiple places along their boundary (as rete #3), will usually be split by the algorithm as if they were series of short retes fused together (the actual split is marked red, whereas the proper split is marked green); (**c**) our method fails to identify the rete base for a group retes fused near the epidermal plate if total width of such a group is too large (as it is the case for the group #2).

## 4. Discussion

In this article we proposed a method for automatic determination of the location, width, length, and height of border irregularities (projections) and demonstrated its application in measuring morphometry of retes in a segmented epidermis.

The main axis of the analyzed region and the approximated location of projections along that axis are determined by analyzing the skeleton of that region obtained using an augmented Fast Marching Method. The main axis is determined by finding such a path in an undirected graph spanned on the skeleton of the region which satisfies a joint criterion of the length and distance from the region's "outer" edge. Points on the region border being candidate nodes for endpoints of projection bases, i.e., points located in centers of border protrusions, are identified by finding prominent peaks the curvature of the region border filtered using a combination of a simple averaging filter and a low-pass filter. Projection roots are then matched with appropriate candidates according to geometric criteria, such as: the relative orientation of a "projection root–candidate node" vector with respect to the orientation of a given projection, and the distance between a node and the root. Since some projections contain multiple tips whereas others are partially joined with their neighboring projections, we perform a post-processing step and either merge or split the affected segmented projections. Finally, we measure the morphometry (width, length and height) of individual projections.

We tested our method on a set of 25 skin whole slide images (WSIs) of common melanocytic lesions. For each image an experienced dermatopathologist manually prepared the ground truth consisting of the epidermis segmentation (the analyzed region) as well as information about individual retes (i.e., downward projections of the epidermis between the underlying connective tissue)—the approximated location of its base endpoints and its length. Experimental results show that the proposed method yields promising results. In total, 825 out of 992 (83%) manually segmented retes were detected correctly and the "strict" Jaccard similarity coefficient for the task of detecting rete ridges was 0.798. Most errors were caused by the misdetection of micro-retes (false negatives) and by treating skin appendages as retes (false positives). The relative error of width and height estimates is on average $\pm 20\%$ and $\pm 16\%$, respectively. Our method has a tendency to rather over- than underestimate values of those morphometric parameters.

In a follow-up study we plan to use the outcome of this method to develop complex indexes describing the epidermal morphometry (variations in epidermal thickness as well as changes in elongation of retes along the lesion) and assess their usefulness for diagnosing skin melanocytic lesions. Pathologists, based on their clinical experience, conjectured that there exist relationships between certain types of distortions and skin conditions, for instance: atrophy of basal epidermal unit is often sighted in melanomas, whereas uniform elongation of retes tends to indicate a benign lesion [7,8].

However, no detailed criteria were formulated so far and we intend to fill this void. We also plan to evaluate the performance of rete segmentation method on automatically segmented epidermis regions. Our method could also help build complex diagnostic systems for melanoma, since it directly allows to verify two criteria for classifying a lesion as a dysplastic nevus (and not melanoma): clubbed-shaped retes and nests of melanocytes confined to the tips of retes.

**Author Contributions:** Conceptualization, P.K. and G.D.; methodology, P.K.; software, P.K.; validation, P.K., G.D. and J.J.-K.; formal analysis, P.K. and J.J.-K.; investigation, P.K.; resources, G.D. and A.G.-J.; data curation, P.K. and A.G.-J.; writing—original draft preparation, P.K.; writing—review and editing, A.G.-J., G.D. and J.J.-K.; visualization, P.K.; supervision, P.K. and J.J.-K.; project administration, P.K.; funding acquisition, P.K.

**Funding:** This work was supported by the National Science Center based on the decision number 2016/23/N/ST7/01361 (to Pawel Kleczek).

**Acknowledgments:** We thank Head of Department of Immunology, Medical University of Warsaw, Jakub Gołąb, for his consent to use the digital whole slide scanner.

**Conflicts of Interest:** The authors declare no conflict of interest.

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
