# Peer review of "A New Approach to Border Irregularity Assessment with Application in Skin Pathology"

_applsci, doi:10.3390/app9102022_

Round 1
Reviewer 1 Report
The authors presented a method to define the epidermis shape which is important for skin cance diagnosis. The paper is well presented and some the results are adequately explained. However some clarifications are needed before the manuscript is published:
Page 2, "These issues may be addressed by developing automatic image analysis method"--> Some dedicated software exist, could you please refer to them and briefly describe what morphometric criteria the use, e.g. I think elongation is one of them.
2. Page 3, first paragraph, --> Stating that the method can be used in other fields like cardiology, is not enough, you have to prove it. Just the statement is not enough.
3. Page 4, The value of tY--> Did you perform some kind of sensitivity analysis for this threshod? Please describe the procedure followed.
4. Page 4, 2.1 last paragraph, --> Then why you describe the segmentation method. You should apply the segmentation and then the rest of the algorithm. Your results should have the error of the the segmentation too. Segmentation is very important to properly detect the normals of the borders.
5. Page 6, threshold t = 70-->how this threshold was decided?
Author Response
We greatly acknowledge the Reviewer for carefully reading the manuscript and providing constructive comments that will lead to an improved paper. We have made changes in the revised manuscript according to your suggestions. Please see below for a description/discussion of how we addressed your major comments.
Page 2, "These issues may be addressed by developing automatic image analysis method"--> Some dedicated software exist, could you please refer to them and briefly describe what morphometric criteria the use, e.g. I think elongation is one of them.
Based on the reviewers suggestion, in the "Introduction" section we have added the "Related works" subsection where we put a list of research works dealing with the subject of epidermis segmentation and computer vision systems for classifying changes. However, none of these systems calculate features associated with epidermal morphometric criteria.
Page 3, first paragraph, --> Stating that the method can be used in other fields like cardiology, is not enough, you have to prove it. Just the statement is not enough.
Thank you very much for pointing out this issue. I fully agree that at the first sight this statement can be misleading and showing potential applications that do not exist. In our work, we describe in detail the process of identifying irregularities, which may appear in various cases, both medical and technical. For each single case/solution the algorithm needs to be refined and adjusted. In our opinion our algorithm can be used in cardiological images including IVUS and OCT. In the future we plan to exploit the implemented method to analyze the above mentioned images which will constitute a separate research area.
We improved the Introduction section by providing the above-mentioned examples from the field of cardiology together with the appropriate references.
Page 4, The value of tY--> Did you perform some kind of sensitivity analysis for this threshold? Please describe the procedure followed.
Thank you very much for your question. Yes, we did measure the mean true positive rate (TPR) and the mean intersection-to-union area ratio (i.e. Jaccard index) of the calculated background mask to the ground-truth mask – for our dataset the simple thresholding with tY = 190 scored mean TPR of 1.0 and JI of 0.992. However, after we tested this approach on WSI images obtained from other sources (taken under different illumination conditions), it turned out that mean TPR and JI dropped significantly – the threshold would require a manual tuning for each new dataset. Therefore, in the revised version of the manuscript we decided to use an adaptive algorithm, where only pixels visually similar to the "average" background color are selected. We described the exact procedure in section "Specimen segmentation" and justified the choice of parameters in section "3.3 Parameter selection". The new adaptive approach scored mean TPR of 0.999 and JI of 0.986 across the diverse set of 12 images. This change of specimen segmentation method did not affect the performance of rete segmentation, as there were no notable differences in the results of the subroutine computing the epidermis outer edge when the new approach was used on our main dataset.
Page 4, 2.1 last paragraph, --> Then why you describe the segmentation method. You should apply the segmentation and then the rest of the algorithm. Your results should have the error of the the segmentation too. Segmentation is very important to properly detect the normals of the borders.
I fully agree that occurring dependencies in the process of creating a CAD system for the healthcare sector is a key issue. In 2017 during the SPIE Medical Conference we have presented our solution for the automated epidermis segmentation in histopathological images of human skin stained with hematoxylin and eosin [1]. The proposed method provides a superior performance compared to the existing techniques. The algorithm segmented the epidermis with a mean sensitivity of 87 %, a mean specificity of 95%. Despite of this, we decided to perform rete ridge analysis on manually segmented images, because the goal of the project was to propose a new method for border irregularity analysis and not to build a computer-aided diagnostic system. In the future we plan to combine individual algorithms and describe achieved results.
[1] Kłeczek, P.; Dyduch, G.; Jaworek-Korjakowska, J.; Tadeusiewicz, R. Automated epidermis segmentation in histopathological images of human skin stained with hematoxylin and eosin. Proc. SPIE 10140, Medical Imaging 2017: Digital Pathology, 101400M, 2017.
Page 6, threshold t = 70-->how this threshold was decided?
Thank you very much for raising this issue. We justified the choice of value for the pruning threshold t in section "3.3 Parameter selection". It turned out that raising that value from 70 to 100 did not affect the performance of our method, as all short boundary details (i.e. 70-100 pixel long) were later rejected due to constraints described in the "Evaluation metrics" section.
Reviewer 2 Report
The manuscript entitled “A New Approach to Border Irregularity Assessment with Application in Skin Pathology” addresses a topic that is interesting, timely and in the scope of the journal.
The manuscript is satisfactory written and organized.
The results/discussion are satisfactory presented and discussed and are convincing.
Therefore, the Reviewer thinks it could be accepted after the following improvements:
- More recent related works could be considered; the following ones are suggested:
a) As image segmentation is one key topic in this article, some review papers related to “segmentation algorithms” would be of great interest for many of its potential readers, including the following examples: 1) “A review on the current segmentation algorithms for medical images”, 1st International Conference on Imaging Theory and Applications (IMAGAPP), ISBN: 978-989-8111-68-5, pp. 135-140, Portugal, 2009; 2) “Segmentation Algorithms for Ear Image Data towards Biomechanical Studies”, Computer Methods in Biomechanics and Biomedical Engineering, 17(8):888-904, 2014; 3) “A Review of Algorithms for Medical Image Segmentation and their Applications to the Female Pelvic Cavity”, Computer Methods in Biomechanics and Biomedical Engineering, 13(2):235-246, 2010; 4) “Segmentation and Simulation of Objects Represented in Images using Physical Principles”, Computer Modeling in Engineering & Sciences, 32(1):45-55, 2008.
b) The following very closed works related to the analysis of skin images should be considered: 1) “A Review of the Quantification and Classification of Pigmented Skin Lesions: From Dedicated to Hand-Held Devices”, Journal of Medical Systems, 39:177, 2015; 2) “Pattern Recognition in Macroscopic and Dermoscopic Images for Skin Lesion Diagnosis”, VipIMAGE 2017, Lecture Notes in Computational Vision and Biomechanics, Vol. 27, ISBN: 978-3-319-68194-8, Springer, DOI: 10.1007/978-3-319-68195-5_55, pp. 504-514, 2018; 3) “From Dermoscopy to Mobile Teledermatology”, Dermoscopy Image Analysis, Chapter 12, pp. 385-418, CRC Press, 2015; 4) “Computational Methods for Pigmented Skin Lesion Classification in Images: Review and Future Trends”, Neural Computing and Applications, 29(3):613-636, 2018; 5) “Social Group Optimization Supported Segmentation and Evaluation of Skin Melanoma Images”, Symmetry, 10(2):51, 2018; 6) “A Computational approach for Detecting Pigmented Skin Lesions in Macroscopic Images”, Expert Systems with Applications, 61:53-63, 2016; 7) “Computational methods for the image segmentation of pigmented skin lesions: A Review”, Computer Methods and Programs in Biomedicine, 131:127-141, 2016; 7) “Computational Diagnosis of Skin Lesions from Dermoscopic Images using Combined Features”, Neural Computing and Applications, DOI: 10.1007/s00521-018-3439-8; 8) “Novel Approach to Segment Skin Lesions in Dermoscopic Images based on a Deformable Model”, IEEE Journal of Biomedical and Health Informatics, 20(2): 615-623, 2016.
- More is needed about: how the values of the used parameters were defined, how they should be defined and what are their influence on the results.
- The limitations of the proposed methodology should be discussed.
Author Response
More recent related works could be considered; the following ones are suggested:
a) As image segmentation is one key topic in this article, some review papers related to “segmentation algorithms” would be of great interest for many of its potential readers, including the following examples: 1) “A review on the current segmentation algorithms for medical images”, 1st International Conference on Imaging Theory and Applications (IMAGAPP), ISBN: 978-989-8111-68-5, pp. 135-140, Portugal, 2009; 2) “Segmentation Algorithms for Ear Image Data towards Biomechanical Studies”, Computer Methods in Biomechanics and Biomedical Engineering, 17(8):888-904, 2014; 3) “A Review of Algorithms for Medical Image Segmentation and their Applications to the Female Pelvic Cavity”, Computer Methods in Biomechanics and Biomedical Engineering, 13(2):235-246, 2010; 4) “Segmentation and Simulation of Objects Represented in Images using Physical Principles”, Computer Modeling in Engineering & Sciences, 32(1):45-55, 2008.
b) The following very closed works related to the analysis of skin images should be considered: 1) “A Review of the Quantification and Classification of Pigmented Skin Lesions: From Dedicated to Hand-Held Devices”, Journal of Medical Systems, 39:177, 2015; 2) “Pattern Recognition in Macroscopic and Dermoscopic Images for Skin Lesion Diagnosis”, VipIMAGE 2017, Lecture Notes in Computational Vision and Biomechanics, Vol. 27, ISBN: 978-3-319-68194-8, Springer, DOI: 10.1007/978-3-319-68195-5_55, pp. 504-514, 2018; 3) “From Dermoscopy to Mobile Teledermatology”, Dermoscopy Image Analysis, Chapter 12, pp. 385-418, CRC Press, 2015; 4) “Computational Methods for Pigmented Skin Lesion Classification in Images: Review and Future Trends”, Neural Computing and Applications, 29(3):613-636, 2018; 5) “Social Group Optimization Supported Segmentation and Evaluation of Skin Melanoma Images”, Symmetry, 10(2):51, 2018; 6) “A Computational approach for Detecting Pigmented Skin Lesions in Macroscopic Images”, Expert Systems with Applications, 61:53-63, 2016; 7) “Computational methods for the image segmentation of pigmented skin lesions: A Review”, Computer Methods and Programs in Biomedicine, 131:127-141, 2016; 7) “Computational Diagnosis of Skin Lesions from Dermoscopic Images using Combined Features”, Neural Computing and Applications, DOI: 10.1007/s00521-018-3439-8; 8) “Novel Approach to Segment Skin Lesions in Dermoscopic Images based on a Deformable Model”, IEEE Journal of Biomedical and Health Informatics, 20(2): 615-623, 2016.
Thank you very much for your suggestion. To make our manuscript more interesting we have analysed the proposed bibliography and added following references to the text:
“A review on the current segmentation algorithms for medical images”, 1st International Conference on Imaging Theory and Applications (IMAGAPP), ISBN: 978-989-8111-68-5, pp. 135-140, Portugal, 2009;
“Segmentation Algorithms for Ear Image Data towards Biomechanical Studies”, Computer Methods in Biomechanics and Biomedical Engineering, 17(8):888-904, 2014;
“A Review of Algorithms for Medical Image Segmentation and their Applications to the Female Pelvic Cavity”, Computer Methods in Biomechanics and Biomedical Engineering, 13(2):235-246, 2010;
“Segmentation and Simulation of Objects Represented in Images using Physical Principles”, Computer Modeling in Engineering & Sciences, 32(1):45-55, 2008.
“A Review of the Quantification and Classification of Pigmented Skin Lesions: From Dedicated to Hand-Held Devices”, Journal of Medical Systems, 39:177, 2015;
“Computational Methods for Pigmented Skin Lesion Classification in Images: Review and Future Trends”, Neural Computing and Applications, 29(3):613-636, 2018;
“Computational methods for the image segmentation of pigmented skin lesions: A Review”, Computer Methods and Programs in Biomedicine, 131:127-141, 2016;
More is needed about: how the values of the used parameters were defined, how they should be defined and what are their influence on the results.
Thank you very much for raising this issue. We justified the choice of values of parameter in section "3.3 Parameter selection". It turned out that raising the valueof the pruning threshold t from 70 to 100 did not affect the performance of our method, as all short boundary details (i.e. 70-100 pixel long) were later rejected due to constraints described in the "Evaluation metrics" section. Although the specimen segmentation using the simple thresholding with tY = 190 worked well on our dataset, it scored mean true positive rate (TPR) of 1.0 and Jaccard index (JI) of 0.992, the threshold would require a manual tuning for each new dataset. Therefore, in the revised version of the manuscript we decided to use an adaptive algorithm, where only pixels visually similar to the "average" background color are selected. We described the exact procedure in section "Specimen segmentation" and justified the choice of parameters in section "3.3 Parameter selection". The new adaptive approach scored mean TPR of 0.999 and JI of 0.986 across the diverse set of 12 images. This change of specimen segmentation method did not affect the performance of rete segmentation, as there were no notable differences in the results of the subroutine computing the epidermis outer edge when the new approach was used on our main dataset.
The limitations of the proposed methodology should be discussed.
Thank you very much for pointing out this issue. We described the three main limitations of our method at the end of section "Performance".
Reviewer 3 Report
The authors proposed a new approach for segmentation in skin pathology. The theory part of the research sounds right, however the following concerns need to be addressed:
There are bunch of typos in the paper and un-defined abbreviations such IQR, ROI, and so on. These need to be defined since the readers might not know by ROI you mean region of interest (I assumed and so on).
Why the authors used Jaccard similarity index? Why the other similarity indices have not been employed? That would be better to add more reference in this regard pointing out to the other researches.
The other point is, I am wondering how machine learning algorithms can evaluate the segmentation results based on the ground truth. It can be done pixel-wise or region-wise classification. Nowadays, deep learning models are being used to even segment noisy images automatically. The authors should prepare a quite exhaustive intro about this section and explain why their method is robust to noise and other/similar methods. The following papers can be added to the paper:
Yuan, Y., Chao, M., & Lo, Y. C. (2017). Automatic skin lesion segmentation using deep fully convolutional networks with jaccard distance. IEEE transactions on medical imaging, 36(9), 1876-1886.
Tahmassebi, A., Gandomi, A., McCann, I., Schulte, M., Goudriaan, A., & Meyer-Baese, A. (2018). Deep learning in medical imaging: fMRI big data analysis via convolutional neural networks. Proc. Pract. Exp. Adv. Res. Comput. ACM.
Jafari, M. H., Karimi, N., Nasr-Esfahani, E., Samavi, S., Soroushmehr, S. M. R., Ward, K., & Najarian, K. (2016, December). Skin lesion segmentation in clinical images using deep learning. In 2016 23rd International conference on pattern recognition (ICPR) (pp. 337-342). IEEE.
Li, Y., & Shen, L. (2018). Skin lesion analysis towards melanoma detection using deep learning network. Sensors, 18(2), 556.
Tahmassebi, A. (2018, May). ideeple: Deep learning in a flash. In Disruptive Technologies in Information Sciences (Vol. 10652, p. 106520S). International Society for Optics and Photonics.
Yuan, Y. (2017). Automatic skin lesion segmentation with fully convolutional-deconvolutional networks. arXiv preprint arXiv:1703.05165.
Author Response
We greatly acknowledge the Reviewer for carefully reading the manuscript and providing constructive comments that will lead to an improved paper. We have made changes in the revised manuscript according to your suggestions. Please see below for a description/discussion of how we addressed your major comments.
There are bunch of typos in the paper and un-defined abbreviations such IQR, ROI, and so on. These need to be defined since the readers might not know by ROI you mean region of interest (I assumed and so on).
Thank you very much for this suggestion. Correction has been made in the revised manuscript.
Why the authors used Jaccard similarity index? Why the other similarity indices have not been employed? That would be better to add more reference in this regard pointing out to the other researches.
Thank you very much for raising this issue. Comparison between two regions or two lines on a binary image is not a trivial task. Many different statistical indexes have been proposed for the analysis of segmented regions, however, due to our knowledge the comparison of lines and sections has not yet been fully elaborated. Based on our experience we propose the Jaccard similarity index in two versions strict and relaxed to compare the manually and automatically drawn lines. This statistical approach has been described in detail on page 14–15. Furthermore, for retes matching exclusively we calculated standard statistical measures describing the distribution of relative errors for both lengths of rete bases and rete lengths. In section "3.4 Performance" the results for individual conditions of the confusion matrix have been described in detail, which allows to calculate any needed statistical metrics.
The other point is, I am wondering how machine learning algorithms can evaluate the segmentation results based on the ground truth. It can be done pixel-wise or region-wise classification. Nowadays, deep learning models are being used to even segment noisy images automatically. The authors should prepare a quite exhaustive intro about this section and explain why their method is robust to noise and other/similar methods.
Deep learning in general and convolutional neural networks in particular have been used in variety of pattern recognition problems like retinal vessel segmentation, lung area detection, or breast cancer classification. In our research, we have recently deployed deep learning methods for vascular structure segmentation and acral melanocytic segmentation [1, 2]. Deep learning methods can solve many difficult and nontrivial tasks in medical image segmentation and classification process. However, at the same time several conditions regarding the database problem statement must be met. For implementing, training and testing a CNN architecture a large database is needed. Furthermore, due to the complexity of the problem the description of individual classes is not explicit. In our opinion, the deep learning algorithms are not recommended for solving the described problem. As shown in the answer to your comment no. 4, we added a few most important deep learning papers to our research article.
[1] Joanna Jaworek-Korjakowska, “A Deep Learning Approach to Vascular Structure Segmentation in Dermoscopy Colour Images,” BioMed Research International, vol. 2018, Article ID 5049390, 8 pages, 2018.
[2] Joanna Jaworek-Korjakowska, "Acral melanocytic lesion segmentation with a convolution neural network (U-Net)," Proc. SPIE 10950, Medical Imaging 2019: Computer-Aided Diagnosis, 109504B (13 March 2019);
The following papers can be added to the paper:
Yuan, Y., Chao, M., & Lo, Y. C. (2017). Automatic skin lesion segmentation using deep fully convolutional networks with jaccard distance. IEEE transactions on medical imaging, 36(9), 1876-1886.
Tahmassebi, A., Gandomi, A., McCann, I., Schulte, M., Goudriaan, A., & Meyer-Baese, A. (2018). Deep learning in medical imaging: fMRI big data analysis via convolutional neural networks. Proc. Pract. Exp. Adv. Res. Comput. ACM.
Jafari, M. H., Karimi, N., Nasr-Esfahani, E., Samavi, S., Soroushmehr, S. M. R., Ward, K., & Najarian, K. (2016, December). Skin lesion segmentation in clinical images using deep learning. In 2016 23rd International conference on pattern recognition (ICPR) (pp. 337-342). IEEE.
Li, Y., & Shen, L. (2018). Skin lesion analysis towards melanoma detection using deep learning network. Sensors, 18(2), 556.
Tahmassebi, A. (2018, May). ideeple: Deep learning in a flash. In Disruptive Technologies in Information Sciences (Vol. 10652, p. 106520S). International Society for Optics and Photonics.
Yuan, Y. (2017). Automatic skin lesion segmentation with fully convolutional-deconvolutional networks. arXiv preprint arXiv:1703.05165.
Thank you very much for your suggestion. To make our manuscript more interesting we have added following references to the text:
Yuan, Y., Chao, M., & Lo, Y. C. (2017). Automatic skin lesion segmentation using deep fully convolutional networks with jaccard distance. IEEE transactions on medical imaging, 36(9), 1876-1886.
Jafari, M. H., Karimi, N., Nasr-Esfahani, E., Samavi, S., Soroushmehr, S. M. R., Ward, K., & Najarian, K. (2016, December). Skin lesion segmentation in clinical images using deep learning. In 2016 23rd International conference on pattern recognition (ICPR) (pp. 337-342). IEEE.
Li, Y., & Shen, L. (2018). Skin lesion analysis towards melanoma detection using deep learning network. Sensors, 18(2), 556.
Round 2
Reviewer 1 Report
I thank the authors for their thorough reply, they have addressed all my comments.
Suggestion - In the Introduction section there are methods (recent) that segment very well the lumen, like:
Optimized Computer-Aided Segmentation and Three-Dimensional Reconstruction Using Intracoronary Optical Coherence Tomography,. Athnasiou, IEEE journal of biomedical and health informatics, 2018
Author Response
Thank you very much for your suggestion, we added the appropriate citation [18] in line 52 (in the bold paragraph in the "Introduction" section).